# The Impact of on-Call Work for Partners' Sleep, Relationship Quality and Daytime Functioning

**Simone M. Karan, Grace E. Vincent, Sally A. Ferguson and Sarah M. Jay ***

CQUniversity, Appleton Institute, Adelaide 5034, South Australia, Australia;
simone.karan@cqumail.com (S.M.K.); g.vincent@cqu.edu.au (G.E.V.); sally.ferguson@cqu.edu.au (S.A.F.)
*   Correspondence: s.jay@cqu.edu.au; Tel.: +61-8-8378-4528

**Abstract:** The on-call schedule is a common work arrangement that allows for the continuance of services during periods of low demand or emergencies. Even though 17%–25% of the world's population participate in on-call work, the human impacts of on-call are generally poorly described in the literature. Of the studies available on the effects of on-call work on workers, disturbances to sleep duration and sleep quality are the most commonly reported, along with negative sleep-related consequences on sleepiness, fatigue, stress and mood. Research has shown that for couples sharing a bed, disturbances to sleep can impair relationship conflict resolution and reduce relationship quality. In the 'off-site' on-call scenario where workers are sleeping at home, their co-sleeping partner may be at risk of sleep disturbances and the subsequent detrimental consequences of this disturbed sleep for themselves and their relationship. To date, few studies have investigated the impact of on-call work for partners' sleep and the potential sleep-related consequences. Therefore, further studies are needed to specifically address whether on-call work impacts the sleep of partners and whether these sleep disturbances also impact the partner's daily performance and relationship quality. Our aim was to provide a narrative around the existing, relevant literature that both investigate and inform the potential impact of on-call for workers' partners' sleep and related consequences.

**Keywords:** on-call; sleep; workers; partners; emotion; relationship quality

---

## 1. Introduction

Non-standard working hours are prolific in our 24-h global society. Not surprisingly therefore, the impacts of various working time arrangements for workers' health and safety have been the focus of investigations which now comprise a significant body of research (e.g., [1–8]). There is also a significant, albeit much smaller body of research that has acknowledged and explored the impacts of these same various working time arrangements for workers' support networks e.g., family, friends and community (e.g., [9–11]). While there are benefits of non-standard working hours, the negative impacts in terms of sleep [1], which plays a pivotal role in health and well-being [12], cannot be understated.

On-call or standby work, is a working time arrangement that requires workers to remain on standby either at a location that is on-site (at work) or off-site (e.g., home), until called to work by their employer [13] and compared to other non-standard working time arrangements its impact is relatively understudied. Further, specific knowledge about the impact of on-call work for workers' family and community are limited to a few, subjective studies, predominantly in the emergency services sector [14–18]. This presents an important gap in current understanding, particularly given that on-call work has significant potential to disturb the sleep of partners and family members due to calls and call-outs during the overnight period in combination with dyadic nature of sleep [19]. Therefore, the aim of this narrative review is to provide information on a) the small body of work that specifically investigates the impacts of on-call work for partners' sleep and b) other, relevant literature that informs

the potential impact of on-call for workers' partners in terms of sleep and subsequent consequences of disturbed sleep such as relationship quality and daytime functioning.

## 2. Methods

There is a scarcity of research that has specifically investigated the impact of on-call work for the partners of the on-call workers in terms of sleep. By necessity, therefore, this review is of a narrative or scoping style rather than systematic or meta-analytical. As noted, the aim of this review is to both evaluate the small body of literature on the sleep of partners of on-call workers and to provide context in terms of dyadic nature of sleep and the known relationship between sleep and waking function, in particular relationship quality. The literature search was conducted as follows.

Regarding the literature on the sleep of partners of on-call workers, the online databases MEDLINE/PubMed and Google Scholar were searched between March 2018–January 2019. The following search terms (individually and in combination) were used "on-call"; "stand-by"; "work", "sleep", "co-sleeping", "disturbance", "quality", "quantity", "partners", "spouses", "dyadic", "relationship", "emotions", "functioning", "regulation", "emergency services", "fire-fighters", "family", "health", "well-being", "alarm", "conflict", "performance", "impact", "consequences".

In addition, the reference lists of original manuscripts were manually searched for other relevant literature. Articles could be either quantitative, qualitative or mixed-method but to be included, had to be written in the English language. In addition, a non-exhaustive search of articles that would inform the finding from the on-call partners' sleep research were also included—overarching themes were the dyadic nature of sleep, the relationship between sleep and waking functioning, emotions and relationship quality.

## 3. The On-Call Context

On-call work arrangements are utilised across a number of work sectors, providing the global community with continuous access to technology, utilities, medical and emergency services, 24 h a day, 7 days a week [15,20]. Current estimates suggest that between 17%–25% of the workforce (Australia, U.S.A and Europe) have an on-call component to their paid or volunteer role [21–23]. On-call work provides a convenient and economical workforce during periods of low or unpredictable service demand, particularly at night, and is common to a number of industries, including healthcare, emergency services, and information technology [20]. However, a disadvantage of on-call work is that work often covers the overnight period and therefore has significant potential to negatively disrupt sleep.

Negative impacts for sleep are the most commonly described consequences of on-call for workers [24], with sleep shown to be impacted even in the absence of actual calls [25,26]. Explanations underlying these disturbances are related to the unpredictability of when or whether a call will be received (i.e., call likelihood), what tasks may be required to be performed upon waking, as well as concern about missing the call [27–30]. These findings have important ramifications for the daily performance of on-call workers as the subjective experience of poor sleep following being on-call has been shown to correspond to increased fatigue and feeling insufficiently recuperated [31]. Such associations appear to be supported by a recent survey study of on-call workers, which revealed that 80% of the Australian on-call population reported experiencing inadequate sleep while on-call compared to when not on-call [32]. This represents a significant number of individuals who may be at risk of adverse health and safety consequences due to poor sleep.

## 4. Sleep as a Shared Behaviour

Though sleep is often assessed in terms of the individual, sleep is also considered a shared, social health behaviour, as reflected by the tendency for people to co-sleep when in a relationship [19]. As summarsied by Troxel at al., (2007), for optimal sleep, the environment in which this behaviour is shared must allow the individuals to feel safe enough to downregulate vigilance and allow the

vulnerable state of sleep to ensue [19]. Further, in the co-sleeping context, sleep is dyadic in nature, meaning that sleep may be influenced by the sleep behaviour of each individual in the co-sleeping pair. For example, in a sample of individuals with obstructive sleep apnoea, it was shown that the regular disturbance associated with obstructive sleep apnoea impacted both the individual and their partner. Further, following treatment, partners' sleep disturbance was significantly reduced and associated with more than 60 min extra sleep per night [33].

In the context of 'off-site' on-call, specifically where workers are sleeping at home, workers' partners are faced with the very real possibility that their sleep will also be disturbed by overnight on-call activity. The next day behavioural outcomes, associated with prior sleep [34,35] are well known but there may also be acute consequences in terms of relationship functioning [36]. On-call investigations to date however, have focused largely on the adverse effects of on-call work on the worker themselves e.g., [25,37–40] and there is very little information on how co-sleeping partners exposed to on-call work are impacted in terms of sleep. Within the existing data, all of which is subjective and limited largely to the emergency services, partners of on-call workers reported that their sleep was disturbed by on-call work and that they also experienced impairment to daily functioning from next day sleep-related consequences such as increased fatigue [16–18].

In line with these reports from partners, on-call workers themselves have expressed concerns that their work arrangement may potentially be even worse for their partners than for the workers [15–17]. For example, on-call workers have reported that the sleep of their partners was directly disturbed when calls occurred overnight, with some partners even waking to answer the phone calls for the on-call worker during the night [41]. In two studies of on-call emergency service workers, the workers reported that their partners often worried about them out on a call [14,15]. Therefore, high risk on-call work as in the emergency services, and the associated worry may present another factor that could reduce partner sleep quantity and quality. Therefore, the impact of on-call work for sleep has implications for both the on-call worker and their partner individually, but may also impact their dynamic and their success as a couple, particularly when the links between sleep and emotional functioning are considered [42].

## 5. Sleep, Emotion and Relationship Quality

Sleep is essential for normal emotional functioning. Poor sleep and consequences of inadequate sleep such as fatigue and sleepiness, have been associated with increased negative emotionality [43], difficulties in regulating emotions and empathic responding to the emotions of others [42,44,45]. Given that emotional regulation is important for maintaining healthy relationships, disruptions to sleep by on-call work may place these relationships at risk of increased negative emotional interactions and conflict [46]. Longitudinal studies of sleep and relationship quality have demonstrated that disruptions to the sleeping environment may influence the function of the overall relationship [46]. For example, poor sleep quality has been reported to contribute to poorer marital quality and may even be a predictor of relationship dissolution [47,48]. In a study of medical interns, work-related sleep loss and resultant fatigue and sleepiness were reported to contribute to decreased ability to communicate and respond empathically to both patients and family members [49].

Studies specifically on co-sleeping echo the above findings, with studies reporting poorer relationship quality and functioning when sleep is inadequate [46,48,50]. In Australia alone, 14.3% of the general population report that sleep disturbances due to the behavior of their partner negatively affect their relationship [33]. Conceivably therefore, any individual whose work pattern has potential to disrupt their sleep, may experience reduced relationship quality as a consequence of either their own sleep disturbance and/or sleep disturbance and associated consequences for their co-sleeping partner. Off-site on-call workers sleeping at home, the focus of this narrative review, are a population that may be vulnerable to these emotional and relationship quality impacts.

## 6. Impact of on-Call for Sleep, Emotion and Relationship Quality

While not specifically addressed in on-call populations—potential impacts of on-call work on relationship quality can be inferred from other studies that have demonstrated that the experience of being on-call corresponds with increased negative mood and irritability and lower desire to socialise and participate in household responsibilities [51]. With the potential for sleep disruption to adversely influence mood and emotional responsiveness, the sleep disturbances due to the on-call work arrangement may have the potential to negatively impact personal relationships.

Interview-based studies with on-call workers have reported that workers find it difficult to juggle their on-call role with personal responsibilities, which were exacerbated by the fatigue and the unpredictable nature of on-call work [14]. Workers have cited that excess sleepiness can interfere with their ability to emotionally engage with their family following periods of on-call activity [41]. Such factors were reported as contributors to work related family conflict in a study involving 113 couples of Australian volunteer emergency service workers [16,17]. This study used structural equation modelling to demonstrate that impacts of the on-call role, including disruption of sleep, significantly contribute to work-family conflict and lowered family relationship quality [16,17].

For auxiliary firefighters and volunteer emergency service workers, who participate in 'offsite' on-call, the on-call component is continuous, where the workers are on-call 24 h a day, 7 days a week, 365 days of the year. Volunteer firefighters listed sleep disturbances as a primary cause for concern for both workers and their partners [16,17] in addition to increased fatigue that interfered with their personal life as well as concerns for the effect the work arrangement had on their partners [15–17]. Along with the potential impacts to sleep and daily performance, on-call work has also been implicated as contributing to increased stress and conflict within the home of workers [16–18,52]. For example, in the case of volunteer firefighters, while the obligation to attend a call is often flexible, and depends on the worker's availability, reports from spouses of workers have indicated that these workers may at times prioritise the on-call role over family and other work, which may also influence spousal and family relationships [16,17].

A study involving the partners of medical practitioners who participated in on-call from home, noted the unpredictable interruptions to sleep and family life were associated with increased family stress, frustration and conflict [18,53]. Similar impacts were noted in a qualitative study that included wives of Canadian firefighters [52]. In this study, family disruptions, for example missed family events and lack of time as a couple due to on-call responsibilities, were common themes among partners of on-call firefighters. In a qualitative study that interviewed the partners of doctors, frustration from sleep disturbances and next day fatigue from overnight callouts were a common theme [18]. Partners also reported that the fatigue symptoms in both themselves and the worker increased household stress, which then interfered with their relationships [18]. Such findings are supported by sleep research that has shown a strong association between increased fatigue from inadequate sleep and poorer emotional functioning [42], which may then impact social interactions [49].

In specific on-call populations, both workers and their partners have described the tendency for partners to take on extra household responsibilities when workers were on-call or when workers were too tired to contribute following a period of on-call activity during the night [16,17,51]. For female partners this may be a significant burden, as females today still take on twice as much of the domestic responsibilities as males, irrespective of the time they spend in employment [18,54]. Partners may therefore be coping with both the effects of having their own sleep disturbed by workers' on-call activity, along with the extra domestic responsibilities placed on them by on-call work impacts on the worker. Such effects have been noted as a source of frustration and family conflict among partners of volunteer on-call emergency service workers in Australia [16,17] and by workers in a French study, where 83% of the workers found the work arrangement to be a burden on their family and to reduce private life satisfaction [41].

## 7. Summary

Work related sleep disturbances contribute to serious effects on health and psychosocial well-being, yet the impacts of such disturbances by on-call work has been poorly defined, compared to other work arrangements, such as shiftwork. While on-call is perceived as a cost-effective strategy for employers, the impact of these work arrangements on the employee and their families is still poorly described in the literature. The majority of the limited literature is qualitative studies of the experience of on-call workers and a few included their partners. Findings from past studies demonstrate that on-call work has a pervasive tendency to impact sleep and negatively affect work performance and relationship quality. Research has also indicated that the partners sharing the sleeping environment of these workers also experience sleep disturbances and the subsequent effects on work performance and relationship quality. Further, partners may face the added burden of taking on extra home responsibilities to cover for workers away or recovering from on-call work.

## 8. Future Research Agenda

Given the potential sleep-related consequences of overnight on-call work for both worker and their co-sleeping partner, the extent to which on-call impacts relationship functioning and other daytime functions is an important consideration for future research. To address this, research is needed to quantify the impact of on-call work on partners who co-sleep with workers participating in off-site arrangements. Based on the literature presented in this narrative review, the impacts for partners' sleep need to be considered in terms of disturbance from actual calls; stress or anxiety when their partner is away on a call (particularly in the case of emergency services or other, high risk work) and disturbance when partner returns home. The inclusion of variables that may moderate the sleep related consequences of on-call worker partners (e.g., age, gender, income, type of work, individual characteristics, health status) will also contribute to a more detailed understanding of the impacts and inform the translation of any new knowledge to workers, their families and workplaces.

Further, research needs to consider also what influence these primary sleep impacts have on secondary outcomes such as behavioural function (sleepiness, performance), relationship quality and functioning, domestic burden and global stress. A clearer understanding of the impacts of on-call work on co-sleeping partners will result in industry that is better placed to support workers and *their* support networks and to develop strategies to mitigate these potential negative consequences and minimise the personal costs of on-call work.

**Author Contributions:** S.M.K.: conceived the idea and wrote the first draft of the manuscript. S.M.J. and G.E.V.: critically evaluated the first draft of the manuscript and wrote the final version of the manuscript. S.A.F.: critically evaluated the final version of the manuscript.

**Funding:** This research received no external funding.

**Acknowledgments:** G.E.V. is supported by an Early Career Fellowship at CQUniversity.

**Conflicts of Interest:** The authors declare no conflict of interest.

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
