# Peer review of "The Impact of on-Call Work for Partners’ Sleep, Relationship Quality and Daytime Functioning"

_2624-5175, doi:10.3390/clockssleep1010016_

Round 1

Reviewer 1 Report

This paper focuses on the impact of on-call work on the sleep and relationship quality of the partner/ spouse. However, the title and the abstract are misleading, as the reader is led to believe that this will be an empirical article. While a sound and comprehensive theoretical review of this topic would be a notable contribution, the authors need to clearly state that that was the intent from the beginning, and in fact, conduct a thorough and comprehensive literature review. As it stands, it appears that the literature review was somewhat arbitrary, and there is no discussion of how certain articles were selected for inclusion while others were not. Furthermore, there is a sizeable literature on shift-work effects on partners which is not addressed here. In summary, while I think this paper addresses an important topic, it is not ready for publication as it does not clearly specify its contribution to the literature, and the methods for conducting the review lack rigor.

Author Response

We thank the reviewer for their comments. Please see the attached file for a point by point explanation.

Reviewer 2 Report

Title: The impact of on-call work for partners’ sleep and relationship quality

I commend the authors for a commentary that is timely, relevant and exceptionally well written.

I have only two minor comments:

1.       Authors may consider adding the functional element into the title; that is, not only are partners’ sleep and relationship quality affected by workers being on call, but also daytime functioning including emotional functioning, fatigue, and household responsibilities, all of which are incorporated into the literature review.

2.       Regarding future research agenda: I would add the need to investigate background variables that may moderate the sleep related consequences of partners of on call workers, such as age, gender, income, type of work (e.g., firefighters vs industry maintenance or healthcare workers), commute characteristics (e.g., distance, mode of transportation), individual characteristics, health status, etc.     

Author Response

(The authors gave the same response as above.)

Reviewer 3 Report

The manuscript covers a literature overview on on-call work and how this working time arrangement affects sleep quality and relationship quality of on-call workers and their partners. The manuscript is well written and I subscribe the call for future research on this understudied, yet demanding working time arrangement. A working time arrangement that even may increase due to the development of new technology, e.g. by making full coverage of night shifts unnecessary. I have some minor comments, in particular concerning the description of on-call work.

Introduction

Page 1, lines 27-29: The authors provide a definition of on-call work and differentiate between on-call work onsite and offsite. Offsite can be differentiated between calls handled at home (e.g. by phone, email or remote access) or workers actually have to go to a specific site. These different kind of on-call arrangements may have a different impact on on-call workers, their partners and relationship quality.  Co-sleeping most probably does not occur onsite. Furthermore, partners probably will worry less about if workers won’t have to leave their home compared to driving to a (remote) worksite. Also the impact on relationship quality may differ whether the worker is at home (offsite) or away (onsite). It is not clear to me if the authors focus on all kinds or a specific on-call arrangement. Please specify and in case the objective of this manuscript is to discuss all kinds of on-call arrangements, I encourage the authors to provide a more precise description which kind of on-call arrangement is discussed in the literature overview.

Page 1, lines 29-30. The author mention on-call industries. Do the authors have examples of typical on-call industries or professions?

Page 1, lines 40-42: Related to the previous comment, do the authors also have information on the prevalence of on-call work, if possible also the distinction between onsite and offsite on-call work, and how often and for how long workers are on average on-call?  Numbers may help in sketching the magnitude of the potential challenges concerning on-call work.

Sleep as shared behavior

Page 2, lines 63-65: “Among these few studies, partners expressed similar concerns as the on‐call workers themselves regarding the negative impacts of on‐call work on their sleep and daily life.” And page  2, lines 70-72: “Partners of on‐call workers reported that their sleep was disturbed by on‐call work and that they also experienced impairment to daily functioning from next day sleep‐related consequences such as increased fatigue [22,23,26].” The authors describe there are no studies examining sleep of partners of on-call workers, yet there seems to be some information of how on-call work affects sleep of partners. This confuses me a little bit. Please elaborate.

Page 2, lines 67-69: “Among these few studies, partners expressed similar concerns as the on‐call workers themselves regarding the negative impacts of on‐call work on their sleep and daily life.” Do these two studies relate the worrying due to being on-call or is it related to the kind of work? I can imagine that there are certain risks in the profession of emergency service workers, irrespective whether they are on-call or on a regular rota.

Sleep, emotion and relationship quality

Page 2, lines 85-90: “Studies specifically on co‐sleeping echo the above findings, with studies reporting poorer relationship quality and functioning when sleep is inadequate [31,33,34]. In Australia alone, 14.3% of the general population report that sleep disturbances due to the behavior of their partner, negatively affect their relationship [35]. In a study of medical interns, work‐related sleep loss and resultant fatigue and sleepiness were reported to contribute to decreased ability to communicate and respond empathically to both patients and family members [36].” The paragraph starts with a study on co-sleeping and ends with a study medical interns. This made me wonder whether medical interns co-sleep during on-call periods? Or do the authors imply the effects of on-call work are prolonged in the period following an on-call period? Please specify, see also my earlier comment on describing the on-call arrangement more precise.

Impact of on-call for sleep, emotion and relationship quality

Page 3, lines 132-134: “For female partners this may be a significant burden, as females today still take on twice as much of the domestic responsibilities as males, irrespective of the time they spend in employment [26,40].” Do the authors have numbers if on-call work is more prevalent among males or females?

Future research agenda

Page 4, lines 157-160: “Based on the literature presented here, the impacts for partners’ sleep need to be considered in terms of disturbance from actual calls; stress or anxiety when their partner is away on a call and disturbance when partner returns home.” The recommendation to focus on stress or anxiety when their partner is away on a call is based on two studies of emergency workers. Can the authors provide a more thorough rationale for this recommendation, as stress or anxiety might also be specific to the profession of emergency services?

Author Response

(The authors gave the same response as above.)

Round 2

Reviewer 3 Report

I complement the authors with a timely and adequat response to the reviewer comments. The vast majority of the initial comments are adequately addressed. I acknowledge that the nature of a narrative review does not need a full methods section (as for a systematic review). Yet, considering the authors now phrase their study as a narrative review, I would expect a short description of the search methods (e.g. database and/or search terms, "snowballing") and/or inclusion criteria (e.g. only English literature, (no) focus on a specific study design, etc.).

Author Response

We thank the reviewer for their careful consideration of our amended manuscript. Please see the attached document for responses to specific queries.

Round 3

Reviewer 3 Report

The authors have sufficienty addressed my comments.